# *Aloe barbadensis* Based Bioactive Edible Film Improved Lipid Stability and Microbial Quality of the Cheese

**DOI:** 10.3390/foods12020229

**Published:** 2023-01-04

**Authors:** Firdous Kouser, Sunil Kumar, Hina F. Bhat, Abdo Hassoun, Alaa El-Din A. Bekhit, Zuhaib F. Bhat

**Affiliations:** 1Division of Livestock Products Technology, SKUAST-Jammu, Jammu 181102, J&K, India; 2Division of Animal Biotechnology, SKUAST-Kashmir, Srinagar 191121, J&K, India; 3Univ. Littoral Côte d’Opale, UMRt 1158 BioEcoAgro, USC ANSES, INRAe, Univ. Artois, Univ. Lille, Univ. Picardie Jules Verne, Univ. Liège, Junia, F-62200 Boulogne-sur-Mer, France; 4Sustainable AgriFoodtech Innovation & Research (SAFIR), F-62000 Arras, France; 5Department of Food Science, University of Otago, 9054 Dunedin, New Zealand

**Keywords:** bioactive film, *A. vera*, kalari, lipid stability, microbial quality, sensory analysis

## Abstract

An attempt was made to develop a bioactive edible film using carrageenan and *A. vera* gel for enhancing the storage quality of cheese using kalari, a popular Himalayan cheese, as a food-model system. The film was evaluated for various physicomechanical and oxidative properties (ABTS (2,2-azino-bis (3-ethylbenzothiazoline-6-sulfonate)) and DPPH (1,1-diphenyl-2-picrylhydrazyl) radical scavenging activities, total flavonoid and phenolic contents). Based on preliminary trials, 1% *A. vera* gel was found to be optimum. The addition of the gel resulted in a significant decrease in moisture content, transparency, solubility, and water-vapor transmission rate and increased the thickness and density of the film. The film showed antimicrobial properties against *E. coli* and significantly (*p* < 0.05) decreased the lipid-oxidation (thiobarbituric acid reactive substances, free-fatty acids, and peroxide values) and increased microbial-quality (total-plate, psychrophilic, and yeast/molds) of the samples during 4-week refrigerated storage (4 ± 1 °C). The film also exhibited a significant positive impact on the sensory quality of the cheese, indicating the potential for commercial applications for quality control of cheese during storage.

## 1. Introduction

Cheese is a popular dairy product that provides high-quality animal proteins and fats in a palatable form [1]. Most of the cheese varieties are a rich source of animal fats and are highly susceptible to lipid oxidation, which can lead to a decline in their quality and nutritive value [2]. While studies have used plant extracts to improve lipid stability and microbial quality, the direct addition of plant extracts to cheese matrices can negatively affect their sensory quality and marketing potential [1]. The use of edible films as a vehicle of the plant extracts for controlled release and slow delivery of bioactive compounds is a novel way of maintaining food quality during storage without compromising the sensory quality [3]. The use of edible and biodegradable films and coatings is a sustainable and environmentally friendly way of packaging foods and has recently received enormous interest from researchers and processors due to the possibility of partial substitution of synthetic and plastic packaging materials [4]. Edible and biodegradable films have been developed using different polysaccharides, proteins, lipids, or composites. Bioactive properties are attributed to adding different food additives, such as plant extracts, antioxidants, or antimicrobials, to the film matrices [5]. These films are generally effective in controlling moisture loss and enhancing the storage quality of foods by reducing adverse chemical reactions, such as lipid and protein oxidation, and improving microbial stability. However, edible and biodegradable films have poor permeability and mechanical properties compared to synthetic plastic films, which is the main barrier to their successful commercial application and acceptance [5].

Studies have developed bioactive edible and biodegradable films using several bioactive molecules and plant extracts for a range of food products; however, only a few studies have reported the development of edible films for dairy products, especially cheese. Therefore, an attempt was made to develop a bioactive edible film to enhance the quality of the cheese during storage. To assess the efficacy and preservative potential of the developed film, kalari, a dry and hard cheese, was used as a food model system. Kalari is a popular cheese variety in the Himalayan region of India. The product is particularly vulnerable to lipid oxidation and rancidity development during storage due to its high-fat content [6]. The film was developed using carrageenan and *A. barbadensis* Miller gel-based powder as a bioactive ingredient. Using natural antioxidants to improve the storage quality of foods has higher consumer acceptance [7,8]. Due to a higher content of phenolics and flavonoids, *A. vera* has been attributed to strong antimicrobial, antioxidant, and radical scavenging properties [9], making it an appropriate candidate for the preparation of bioactive films for food packaging. Several studies have reported the chemical characterization and composition of the *A. barbadensis* Miller gel [10,11]. It contains organic acids, such as cinnamic, chrysophanic, acetic, salicylic, lactic, and succinic acid; phenolic compounds, such as anthraquinones, and other phytochemicals responsible for its antimicrobial, antifungal, and antioxidant properties [12]. The gel of the plant has been reported to have antimicrobial properties against Gram-positive, Gram-negative, and opportunistic microbes, including *E. coli*, *S. aureus*, *P. fluorescens*, *P. aeruginosa*, *B. cereus*, *K. pneumonia*, *A. niger* and *C. albicans* [12]. The *A. vera* and carrageenan-based film were developed, characterized, and evaluated for its efficacy using samples of kalari during refrigerated storage for 4 weeks.

## 2. Materials and Methods

### 2.1. Chemicals and Media

The analytical grade chemicals viz. aluminum chloride (AlCl_3_), carrageenan, 2,2′-azino-bis (3-ethylbenzothiazoline-6-sulfonic acid) diammonium salt (ABTS), gallic acid, quercetin, 2,2-diphenyl-1-(2,4,6-trinitrophenyl) hydrazyl (DPPH), 1,1,3,3 tetra-ethoxypropane, and Folin–Ciocalteu’s phenol reagent were purchased from Sigma-Aldrich (Bangalore, India). Media (readymade) for microbial studies and glycerol were purchased from Hi-Media (Mumbai, India). All the other chemicals and reagents used in the study were of analytical or food grade and were purchased from standard suppliers in India, such as Qualigens, Merck, and Sigma-Aldrich.

### 2.2. Preparation of the Film and A. vera Extract

The *A. vera* gel used for the preparation of the extract was harvested from the leaves obtained from *A. vera* plants (*A. barbadensis* Miller) cultivated on the farm at the campus of the university (SKUAST-J, J&K, Jammu, India). After washing the leaves with potable water, the gel fillets (parenchymatous tissue within the leaves) inside the rind were removed from the leaves using a sterile knife. The *A. vera* latex (bitter yellow exudate) was allowed to drain from the fillets before use, and the fillets were homogenized using a domestic blender. The homogenate was not pasteurized to avoid any loss of activity and was frozen at −80 °C for 48 h and was freeze-dried in a freeze drier (INOFD-12S, Innova Bio-Meditech Inc., Boston, MA, USA) for 24 h to obtain a powder. The gel was evaluated for both antioxidant (ABTS and DPPH radical scavenging activities, total flavonoid and phenolic contents) and microbial (total plate count, coliforms, minimum inhibitory concentration against *E. coli* (MIH)) quality before use. These results were used to decide the concentration levels used for the development of the film.

The method followed by Mahajan et al. [13] was used to develop the film, and different levels of glycerol, carrageenan and *A. vera* gel powder were used to optimize the film. Based on the results of the preliminary trials, 14% glycerol, 1.5% carrageenan, and 1% *A. vera* gel were found to be optimum for development of a desirable film. Glycerol (14%) was mixed with the aqueous solution of carrageenan (1.5% solution prepared by stirring on a magnetic stirrer (Glassco laboratory Pvt. Ltd., Ambala Cantt, Haryana, India) at 350 rpm for 20 min at 80 °C) and the solution was stirred for 10 min. After this, *A. vera* powder was added, and the mix was stirred for 10 min and allowed to cool at 60 °C. The mix was cast on glass plates (10 × 20 cm). A hot air oven was used to dry the films (55 °C, 5 h).

### 2.3. Preparation of the Kalari

The method of Bukhari et al. [14] was followed to prepare the kalari. Buffalo milk was standardized (5.5% fat, 9% solids-not-fat) and used for the development of the cheese. Pearson’s square method of milk standardization was used to standardize the whole fresh buffalo milk (6% fat) using buffalo skim milk. The process involved heating milk to 40 °C, followed by its coagulation using 5% lactic acid as the coagulant. The curd obtained after draining the whey through a muslin cloth was turned into small balls (~70 g), which were immediately molded into disc-shaped raw kalari using petri plates as molds. The products were air dried to 30% moisture using traditional bamboo baskets at room temperature and were turned after every 5 h for 3 days. The products were stored in the refrigerator for four weeks, and the samples packaged within the films containing *A. vera* (T2) were compared with samples stored within control films (T1 = film without *A. vera*) and without film (C). The samples were compared for lipid stability, microbial growth, physicochemical characteristics, and sensory quality on days 0, 14, and 28.

### 2.4. Physico-Mechanical Characteristics

The film samples were compared for various physicomechanical characteristics to evaluate the impact of the extract. Parameters such as thickness (mm), moisture content (%), solubility (%), transparency (absorbance value), density, and water vapor transmission rate (WVTR) were evaluated using the methods followed by Mahajan et al. [13]. A micrometer (mm) (Swastik Scientific Company, Mumbai, India) was used to determine the thickness of the film and was analyzed at eight different locations for each film. A hot air oven (Narang Scientific Works, New Delhi, India) was used to determine the moisture percent by gravimetric method. The small pieces (2 × 3 cm) of the film were dried at 100 °C for 24 h, and the percent weight loss was expressed. The samples dried in an oven (2 × 3 cm) for moisture content were weighed, mixed with distilled water, and stirred gently for 2 min. The samples were again dried in an oven at 100 °C till a stable weight reading was obtained. The solubility of the film was measured as the weight difference after and before dissolution was expressed on a percentage basis. Six measurements were taken for each film to determine the transparency. They were measured by the ratio of absorbance at 550 nm to that of the thickness of a rectangular piece (3 cm × 1 cm) of a film placed in a water-containing cuvette.

The floatation method was used to determine the density of the film (1.5 × 1.5 cm) using carbon tetrachloride (1.5935 g/mL) and heptane (0.71 g/mL) as solvents. After immersing the film in 5 mL heptane in a beaker, carbon tetrachloride was added dropwise from a burette until the film floated in the middle. The density (g/mL) was measured using the formula (V1d1 + V2d2/V1 + V2), where V1 and V2 are the volume of heptane and CCl_4_ in ml, and d1 and d2 are the density of heptane and CCl_4_ in g/mL. The WVTR was determined using a test cell containing distilled water (15 mL), sealed with a film, and kept in a desiccator for 24 h. The weight loss from the test cell (WVTR) was measured following the equation {ΔW/(Δt × A)}, where ΔW was the weight loss, A was the area of film exposed, and Δt was the time of storage.

### 2.5. Antioxidant Potential

#### 2.5.1. Total Flavonoid and Total Phenolic Contents

The methods used by Mahajan et al. [13] were followed for determining total flavonoid and total phenolic contents using aluminum chloride calorimetric assay and the Folin–Ciocalteu method, respectively. The test samples (0.5 mL) were mixed with 2% AlCl_3_ ethanol solution (0.5 mL) and were kept at room temperature for 1 h. The absorbance of the mixture was measured at 420 nm using a UV-Vis spectrophotometer (Bengaluru, India, Systronics). Quercetin was used as the standard, and the results were presented in terms of mg Quercetin equivalents (QE)/g. For total phenolics, gallic acid prepared in ethanol was used as the standard. Then, 20 μL of Folin–Ciocalteu reagent was mixed with aliquots (40 μL) of standard or samples and were incubated for 3–5 min at 25 °C. After this, NaCO_3_ solution (20%) was added to the samples and incubated for 30 min. The absorbance was measured (700 nm), and the results were presented in terms of mg Gallic acid equivalents (GAE)/g.

#### 2.5.2. ABTS and DPPH Radical Scavenging Activities

The potential of the samples to scavenge ABTS and DPPH radicals was determined using the methods of Mahajan et al. [13]. A solution of ABTS of 7 mM strength was prepared in water. The mixture of 2.45 mM potassium persulfate and ABTS solution (1:1 *v*/*v*) was kept for 12–16 h in the dark at room temperature to produce the ABTS + radicals (ABTS + solution) before use. The ABTS+ solution was then diluted with distilled water till the absorbance was reduced to 0.70 ± 0.02 (734 nm). The mixture of the 40 μL test sample and the diluted ABTS + solution (5 mL) was placed in the dark (6 min), and the samples of ultra-pure water were used as a blank. The results of the samples were expressed using Equation (1).
(Ab − As)/Ab × 100 = ABTS radical scavenging activity (%)(1)
where Ab and As represent the absorbance of the blank and sample, respectively.

The absorbance of the DPPH working solution and reaction mixture was measured at 517 nm. The reaction mixture contained a methanolic extract of the samples (1 mL) and 0.1 mM DPPH solution (2 mL) in test tubes and was allowed to stand in the dark for 30 min. The results of the samples were expressed using Equation (2).
(Ao − A1)/Ao × 100 = DPPH radical scavenging activity (%)(2)
where Ao and A1 represent the absorbance of the DPPH solution and reaction mixture, respectively.

### 2.6. Microbiological Analysis

#### 2.6.1. Microbial Counts

The methods described by APHA [15] were followed to enumerate (log_10_ cfu/g) various microbiological counts, viz. yeast/molds, coliform, total plate, and psychrophilic, respectively, using potato dextrose agar, violet-red bile agar, and total plate agar. The samples were prepared within the laminar flow (Thermo Electron Corporation, Langenselbold, Germany) near a flame, and the spread plate technique was used to inoculate the samples. Only the results of those plates that contained 30–300 colonies were considered.

#### 2.6.2. Minimum Inhibitory Concentration (MIC) and Disc Agar Diffusion Method

The MIC and the disc agar diffusion tests were determined against *E. coli* (10^6^ cfu/mL) using Muller Hinton broth and Muller Hinton agar, respectively. A bacteriological incubator (Macro Scientific Works, Delhi, India) was used to incubate the samples for 24 h at 37 °C as in [16]. The discs cut out from the films of 10 mm diameter were used, and a digital vernier caliper was used to measure the inhibitory halos in mm. Among bacteria, *E. coli*, *Bacillus subtilis*, and *Staphylococcus aureus* strains are frequently used to determine the antimicrobial potential of plant extracts and identify antibacterial compounds [17]. *E. coli* is a common inhabitant of animal and human gut. It is found in soil, water, and vegetation and is a common cause of human food and water-borne diseases worldwide [18]. The *E. coli* used in the study was isolated from a water source and was maintained in the laboratory on nutrient broth glycerol stock (25%). For the experiment, the *E. coli* was revived using nutrient broth, and characteristics were checked by cultural growth followed by biochemical testing.

### 2.7. pH, Moisture (%), and Lipid Oxidation

The pH and moisture content were measured following the methods described by [19,20] using a digital pH meter (IKA labor Technik, Janke and Kenkel, Breisgau, Germany), Ultra Turrex T10 tissue homogenizer (Cole Parmar, Mumbai, India), and a hot air oven. The lipid stability of the cheese samples was measured using different parameters, i.e., peroxide value (PV), free fatty acids (FFA), and thiobarbituric acid reactive substances (TBARS), using the methods followed by Mahajan et al. [13]. The results were expressed as meq/kg, % oleic acid, and mg malonaldehyde/kg, respectively. A standard (1,1,3,3 tetra-ethoxypropane) was used for measuring the TBARS values.

### 2.8. Sensory Analysis

The sensory analysis of the cheese was performed following an 8-point descriptive scale using a trained panel composed of faculty members and postgraduate students. The 10-member panel comprised both genders (5 females and 5 males), and the age group ranged from 25 to 50 years. After explaining the product attributes, the panelists were provided with three-digit codes and potable water samples for cleansing the palate. The experiments were conducted according to established ethical guidelines, and the study complied with all the regulations. Informed consent was obtained from the sensory participants. The sensory analysis was voluntary; no incentive was given, and the panelists could withdraw from the evaluation at any time without giving a reason. The products evaluated were safe for consumption. The panel was trained for four basic tastes, i.e., recognition and threshold tests and hedonic tests routinely performed in the division.

### 2.9. Statistical Analysis

The data was collected and analyzed using Statistical Package for Social Sciences (SPSS) version 21.0 (SPSS Inc., Chicago, IL, USA). The results are presented in the figures and tables as means ± standard errors. The experiments were replicated six times (*n* = 6 for each treatment) except for sensory analysis, which was performed by ten trained panelists thrice for each treatment. The effect of packaging and storage time was evaluated using DMRT (Duncan’s multiple range test) at a 0.05 significance level. One-way ANOVA and *t*-test were used to analyze the film characteristics. In contrast, the data obtained from storage studies and sensory analysis were analyzed using 2-way ANOVA and repeated measurements ANOVA, respectively.

## 3. Results and Discussion

### 3.1. Physico-Mechanical Properties

Figure 1 and Figure 2 present the data of various physicomechanical parameters of the *A. vera*-based edible films.

The addition of *A. vera* decreased (*p* < 0.05) the transparency (absorbance value), solubility (%), moisture content (%), and water vapor transmission rate (WVTR, mg/m^2^t), however, the density (g/mL) and thickness (mm) of the film exhibited a significant increase (*p* < 0.05). Adding ingredients to the film formulation can alter the composition and affect the physicomechanical properties [21]. The presence of polyphenols and flavonoids in *A. vera* can change the rheological properties of the film material through interactions with proteins and polysaccharides and the formation of linkages [22]. *A. vera* extract has been found to have a higher content of phenolics and flavonoids [9]. By increasing the density and thickness of the film, these changes led to a significant decrease in the transparency and WVTR. The physicomechanical parameters, such as density and water vapor transmission rate, have high significance and can affect the evaporative moisture loss and sensory quality of food products during storage. Increased density and lower water vapor transmission rate of the *A. vera*-based film allowed a lower moisture loss from the stored samples. This was reflected in the moisture content (higher values) and juiciness scores of the cheese samples. The polyphenolic compounds can interact with hydrocolloids and affect their structure, particle size, and hydrophobicity/hydrophilicity and subsequently affect the moisture content, solubility, and distribution of the film-forming solutions [21,23]. Studies have reported similar effects of polyphenol-rich extracts on bioactive films developed for other food products. Both Mahajan et al. [13] and Saricaoglu et al. [23] have reported a significant increase in the density and a decrease in the WVTR of edible films containing the plant extracts.

### 3.2. Antioxidant Potential of the Films

The antioxidant potential of the film was evaluated during the storage (28 days) by determining the total phenolic content (TPC, mg Gallic acid equivalents (GAE)/g), total flavonoid content (TFC, mg Quercetin equivalents (QE)/g) and DPPH and ABTS radical scavenging activities (% inhibition) (Table 1). The TPC, TFC, DPPH, and ABTS radical scavenging activities of the *Aloe vera* gel were 50.85 ± 2.58, 14.29 ± 1.19, 51.02 ± 0.89, and 64.89 ± 0.439, respectively. The *A. vera*-based film containing 1.0% extract showed significantly (*p* < 0.05) higher values for all these antioxidant parameters compared to the control film during the entire storage period. The *A. vera* gel contains numerous phytochemicals such as lectins, terpenoids, flavonoids, fatty acids, anthraquinones (such as aloetic acid, aloin A and B, aloe-emodin, anthranol, emodin, isobarbaloin, ester of cinnamic acid), tannins, polysaccharides, sterols, salicylic acid, enzymes (such as superoxide dismutase, catalase, and glutathione peroxidase), minerals, and vitamins [24]. Many of these phytochemicals have antioxidant properties such as phenolic compounds, α-tocopherol, ascorbic acid, organic acids, carotenoids, and tannins, which might be responsible for the strong antioxidant potential of the films [25]. Other phytochemicals such as saponins (damage microbial DNA and RNA), tannins (inactivates adhesins of bacteria), flavonoids (induce lysis and inhibit cell wall formation), and anthraquinones have strong antimicrobial properties and can kill microbes or inhibit their growth [25].

Previous studies have found a strong antioxidant potential in *A. vera* gel extract [24]. An ethanolic gel extract (95%) of *A. vera* was reported to have a TPC of 413 ± 9.88 mg/100 g GAE and a TFC of 33.6 ± 1.98 mg/100 g CE (catechin equivalents) [26]. Since the phenolic compounds mainly scavenge free radicals produced during the oxidation of proteins and lipids, the estimation of TPC gives a direct estimate of the antioxidant potential of the plant extracts. A linear correlation has been established between total phenolics and radical scavenging activity [27]. A higher value of TPC is always associated with a higher content of flavonoids [28]. The *A. vera* gel has been reported to scavenge the DPPH and ABTS radicals in a dose-dependent manner with IC_50_ values of 572.14 and 105.26 μg/mL, respectively [29]. The higher antioxidant potential of the *A. vera*-based film indicates its potential to control the lipid oxidation of the foods during storage and was confirmed by the results of the lipid stability.

### 3.3. Lipid Stability of the Cheese Samples

The lipid stability of the stored samples was determined using TBARS, FFA, and PV (Table 2). No difference (*p* > 0.05) was found in the lipid stability of the samples on day 0, whereas significantly (*p* < 0.05) lower values were found for TBARS, FFA, and PV for the samples stored within the *A. vera* films (T2) compared to the control (C) and T1 films on days 14 and 28. All these parameters are indicators of freshness and oxidative damage during storage and are commonly used for evaluating the quality of food products during storage. Several environmental and food factors, such as packaging systems, time of storage, moisture content, the concentration of metal ions, and the presence of antioxidants and lipases in the food matrices, play a crucial role in lipid oxidation [3]. Adding plant extracts rich in polyphenols is a convenient and effective way to enhance the lipid stability of food products by directly adding to the food matrices or applying bioactive films and packaging systems [30]. The antioxidant and polyphenol-enriched bioactive films are novel preservative systems that deliver only a required amount of phytochemicals to induce a preservative effect while avoiding significant damage to sensory quality. These films also provide a physical barrier between food products and their environment and help in reducing evaporative moisture loss and lipid oxidation [31]. Higher lipid stability of the cheese samples stored within the films containing *A. vera* indicates their potential to maintain the sensorial quality of food during storage. The sensory analysis results confirmed this result. Previous studies have found a positive effect of plant extract-based bioactive films on the lipid oxidation of milk products, such as cheese, during storage [30,31].

### 3.4. Microbiological Characteristics

#### 3.4.1. Disc Agar Diffusion Test

The addition of *A. vera* imparted antimicrobial properties to the edible film (T2). The size of the inhibitory halos against *E. coli* was significantly (*p* < 0.05) larger for the *A. vera*-based films (Figure 2B). Studies have observed the strong antimicrobial properties of *A. vera* extract against various pathogenic and opportunistic microbes, including *E. coli* [32].

#### 3.4.2. Minimum Inhibitory Concentration (MIC)

The MIC was estimated to determine the antimicrobial potential of the extract powder against *E. coli*, a pathogenic microbe that can grow at lower temperatures [33]. A MIC value of 0.50% showed the antimicrobial properties of the extract. These results agreed with the findings of published studies such as Mahajan et al. [13] and Prasad et al. [34], who observed antimicrobial properties for *A. vera* gel and reported a MIC value of 1.87% and 1.80% against *E. coli* and *E. faecalis*, respectively.

#### 3.4.3. Microbial Counts

A significant effect of the film was observed on the microbial growth of cheese samples during storage (Table 3). In comparison, no difference was found in the microbial counts of the samples on day 0; significantly (*p* < 0.05) lower counts were found for total plate, yeast/molds, and psychrophiles for the samples stored within the *A. vera* films (T2) compared to the control (C) and T1 films on days 14 and 28. The *A. vera* gel contains several antimicrobial compounds, such as coumaric acid, pyrocatechol, and cinnamic acid, and has proven activities against severe spoilage and pathogenic microbes such as *S. marcescens, E. faecalis, E. aerogenes, P. aeruginosa, E. coli, S. agalactiae, K. pneumoniae, S. aureus, S. pyogenes,* and *B. subtilis* [35,36]. The mean values of the microbial counts of all the cheese samples were within the food safety limits after four weeks of storage. The yeast/molds and psychrophiles were detected after two weeks. In contrast, coliforms were not detected for up to four weeks of storage time due to the hygienic practices followed during the processing and sterile packaging. The generic coliform test indicates unsanitary manufacturing conditions in cheese production [37].

### 3.5. pH and Moisture Content

The mean pH and moisture content of the cheese samples during storage are presented in Table 3. The pH of the kalari cheese stored within the *A. vera* films (T2) was significantly (*p* < 0.05) lower than the cheese packaged within the T1 films and control samples on days 14 and 28. The *A. vera* gel is a good source of organic acids, such as cinnamic acids, caffeic acid, chlorogenic acids, and feruloylquinic acid, which can reduce pH [38]. A lower pH is a favorable change that can increase the microbial quality of cheese samples during storage. Studies have found a reduction in the pH of the food samples stored within bioactive edible and biodegradable films containing different phytochemicals [3]. A significant (*p* < 0.05) effect of the film was also recorded on the evaporative loss of moisture from the stored samples. The moisture content of the cheese stored within the films (T1 and T2) was significantly higher than the control cheese on days 14 and 28. Edible and biodegradable film packages are a physical barrier between the food surface and its environment and reduce the loss of moisture from the product surface [30,31].

### 3.6. Sensory Analysis

The effect of the film was also evaluated on the sensory attributes of the cheese during storage (Table 4). While no difference was found in the sensory scores of the cheese on day 0, significantly (*p* < 0.05) higher scores were found for all sensory attributes for the cheese stored within the T2 films compared to control (C) and T1 films on days 14 and 28.

The higher storage stability of the cheese stored within the T2 films improved their sensory quality. Sensory attributes, such as flavor, color, and texture, are significantly affected by the secondary and primary metabolites and free fatty acids released during lipid and protein oxidation of the food products during storage [7]. By reducing the loss of moisture from the packaged samples, the films increased the moisture content and juiciness scores of the samples [23]. The phenolics and other antioxidant compounds positively impact color retention and reduce the production of metabolic compounds, which can affect the color of the food products during storage [31]. The application of films containing bioactive phytochemicals has been found to improve the sensory quality of food products, including cheese, by reducing oxidative and microbial alterations and, consequently, the production of metabolites and off-flavor compounds [39,40]. While glycolysis, proteolysis, and lipolysis are principal pathways for the production of flavor compounds in cheese, lipid oxidation of unsaturated fatty acids and microbial spoilage can produce several odorants and volatile and bitter compounds, such as hydrophobic peptides, octanal, nonanal, hexanal, 2-octenal, decanal, 2-decenal, and 2-undecenal [40].

## 4. Conclusions

The carrageenan and *A. vera* gel extract-based film was developed to improve the storage stability of cheese. The film was characterized for physicomechanical parameters and evaluated for antimicrobial and antioxidant potential. The addition of the extract (1.0%) significantly enhanced the film’s antioxidant potential and antimicrobial properties. Packaging the cheese samples within the film enhanced the lipid stability (TBARS, FFA, and PV) and reduced the microbial growth (total plate, psychrophilic, and yeast/mold counts) during storage. Significantly higher scores were found for all the sensory attributes of the cheese stored within the film. The bioactive film can be used for commercial applications to improve the quality of the cheese during storage.

## Figures and Tables

**Figure 1 foods-12-00229-f001:**
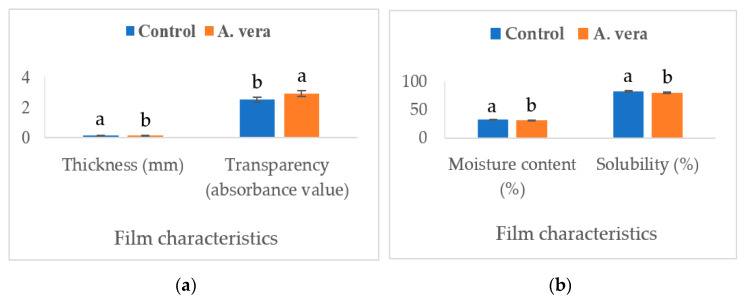
Effect of *A. vera* gel on the film characteristics ((**a**). thickness and transparency, (**b**). moisture content and solubility). (Different superscripts on columns for a parameter differ significantly, *n* = 6, one-way ANOVA was used at a 0.05 level of significance, Control = control film without *A. vera, A. vera* = film containing 1.0% *A. vera*).

**Figure 2 foods-12-00229-f002:**
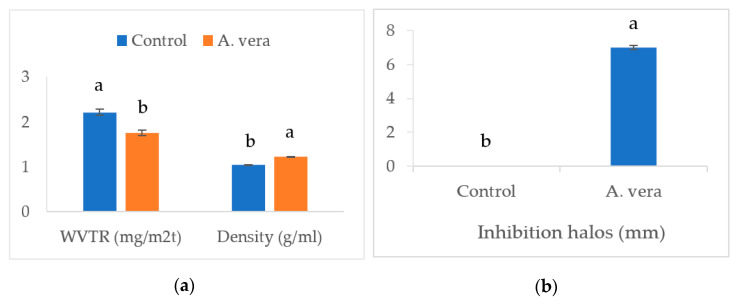
Effect of *A. vera* gel on the film characteristics (**a**) and antimicrobial potential against *E. coli* (**b**). (Different superscripts on columns for a parameter differ significantly, *n* = 6, one way ANOVA was used at 0.05 level of significance, WVTR = water vapor transmission rate, Control = control film without *A. vera*, *A. vera* = film containing 1.0% *A. vera*).

**Table 1 foods-12-00229-t001:** Effect of *A. vera* gel on the antioxidant potential of the film.

Treatments	Storage Period (Days)		
	0	14	28
Total phenolic content [Gallic acid equivalents (mg/g)]			
T_1_	2.88 ± 0.12 ^Ac^	1.20 ± 0.14 ^Ab^	0.84 ± 0.16 ^Aa^
T_2_	39.93 ± 1.04 ^Bc^	25.86 ± 1.02 ^Bb^	20.50 ± 1.06 ^Ba^
Total flavonoid content [Quercetin equivalents (mg/g)]			
T_1_	0.19 ± 0.78 ^Ac^	0.13 ± 0.75 ^Ab^	0.05 ± 0.70 ^Aa^
T_2_	11.71 ± 1.60 ^Bc^	6.73 ± 1.65 ^Bb^	4.34 ± 1.55 ^Ba^
DPPH radical scavenging activity (%)			
T_1_	3.56 ± 0.34 ^Ac^	1.03 ± 0.08 ^Ab^	1.07 ± 0.01 ^Aa^
T_2_	34.64 ± 0.42 ^BBC^	26.20 ± 0.40 ^Bb^	20.47 ± 0.04 ^Ba^
ABTS radical scavenging activity (%)			
T_1_	4.24 ± 0.55 ^Ab^	3.20 ± 0.39 ^Aab^	2.04 ± 0.44 ^Aa^
T_2_	40.56 ± 0.40 ^Bc^	35.48 ± 0.64 ^Bb^	25.3 ± 0.59 ^Ba^

Mean ± SE with different superscripts in a row (lower case alphabet) and column (upper case alphabet) differ significantly (*p* < 0.05). Data were analyzed using two-way ANOVA, *n* = 6 for each treatment. T_1_ = control film without *A. vera*. T_2_ = film containing 1.0% *A. vera* extract.

**Table 2 foods-12-00229-t002:** Effect of *A. vera* gel-based film on the lipid stability of the cheese during refrigerated storage.

Treatments	Storage Period (Days)		
	0	14	28
FFA (% oleic acid)			
C	0.0571 ± 0.009 ^a^	0.0721 ± 0.002 ^Bb^	0.0984 ± 0.001 ^Bc^
T_1_	0.0567 ± 0.008 ^a^	0.0715 ± 0.004 ^Bb^	0.0974 ± 0.006 ^Bc^
T_2_	0.0564 ± 0.007 ^a^	0.0690 ± 0.004 ^Ab^	0.0880 ± 0.002 ^Ac^
TBARS (mg malondialdehyde/kg)			
C	0.272 ± 0.016 ^a^	0.670 ± 0.012 ^Bb^	1.113 ± 0.013 ^Bc^
T_1_	0.267 ± 0.012 ^a^	0.660 ± 0.015 ^Bb^	1.105 ± 0.017 ^Bc^
T_2_	0.264 ± 0.008 ^a^	0.570 ± 0.010 ^Ab^	0.980 ± 0.009 ^Ac^
Peroxide value (meq/kg)			
C	0.214 ± 0.008 ^a^	0.480 ± 0.004 ^Bb^	0.589 ± 0.007 ^Bc^
T_1_	0.210 ± 0.013 ^a^	0.472 ± 0.010 ^Bb^	0.580 ± 0.016 ^Bc^
T_2_	0.208 ± 0.017 ^a^	0.380 ± 0.006 ^Ab^	0.490 ± 0.012 ^Ac^

Mean ± SE with different superscripts in a row (lower case alphabet) and column (upper case alphabet) differ significantly (*p* < 0.05). Data were analyzed using two-way ANOVA, *n* = 6 for each treatment. C = samples without film. T_1_ = samples with control film without *A. vera*. T_2_ = samples with a film containing 1.0% *A. vera* extract.

**Table 3 foods-12-00229-t003:** Effect of *A. vera* gel-based film on the microbiological stability of the cheese during refrigerated storage.

Treatments	Storage Period (Days)		
	0	14	28
Total plate count (log_10_ cfu/g)			
C	2.28 ± 0.024 ^a^	3.45 ± 0.030 ^Bb^	3.60 ± 0.021 ^Bc^
T_1_	2.26 ± 0.034 ^a^	3.36 ± 0.040 ^Bb^	3.54 ± 0.028 ^Bc^
T_2_	2.23 ± 0.036 ^a^	2.50 ± 0.039 ^Ab^	2.70 ± 0.024 ^Ac^
Psychrophilic count (log_10_ cfu/g)			
C	ND	2.50 ± 0.012 ^Ba^	2.80 ± 0.014 ^Bb^
T_1_	ND	2.43 ± 0.020 ^Ba^	2.75 ± 0.011 ^Bb^
T_2_	ND	1.70 ± 0.018 ^Aa^	1.85 ± 0.09 ^Ab^
Yeast and mold count (log_10_ cfu/g)			
C	ND	1.92 ± 0.057 ^Ba^	2.80 ± 0.036 ^Bb^
T_1_	ND	1.84 ± 0.050 ^Ba^	2.74 ± 0.035 ^Bb^
T_2_	ND	0.84 ± 0.049 ^Aa^	1.80 ± 0.034 ^Ab^
pH			
C	4. 96 ± 0.023 ^c^	4.80 ± 0.018 ^Bb^	5.40 ± 0.020 ^Ba^
T_1_	4.90 ± 0.016 ^c^	4.75 ± 0.023 ^Bb^	5.34 ± 0.025 ^Ba^
T_2_	4.89 ± 0.018 ^c^	4.55 ± 0.022 ^Ab^	5.20 ± 0.024 ^Aa^
Moisture (%)			
C	32.14 ± 0.028 ^a^	31.40 ± 0.030 ^Bb^	30.30 ± 0.035 ^Bc^
T_1_	32.18 ± 0.025 ^a^	31.50 ± 0.028 ^Ab^	30.40 ± 0.030 ^Ac^
T_2_	32.21 ± 0.035 ^a^	31.63 ± 0.032 ^Ab^	30.52 ± 0.028 ^Ac^

Mean ± SE with different superscripts in a row (lower case alphabet) and column (upper case alphabet) differ significantly (*p* < 0.05). Data were analyzed using two-way ANOVA, *n* = 6 for each treatment. ND = not detected (Detection limit < 10 cfu/g), Coliforms were not detected during the entire storage time. C = samples without film, T_1_ = samples with control film without *A. vera,* T_2_ = samples with a film containing 1.0% *A. vera* extract.

**Table 4 foods-12-00229-t004:** Effect of *A. vera* gel-based film on the sensory quality of the cheese during refrigerated storage.

Treatments	Storage Period (Days)		
	0	14	28
Color and appearance			
C	7.50 ± 0.067 ^c^	6.40 ± 0.059 ^Bb^	5.50 ± 0.042 ^Ba^
T_1_	7.55 ± 0.042 ^c^	6.48 ± 0.040 ^Bb^	5.57 ± 0.036 ^Ba^
T_2_	7.59 ± 0.050 ^c^	7.20 ± 0.055 ^Ab^	6.50 ± 0.059 ^Aa^
Flavor			
C	7.60 ± 0.043 ^c^	6.60 ± 0.048 ^Bb^	5.60 ± 0.039 ^Ba^
T_1_	7.64 ± 0.044 ^c^	6.68 ± 0.040 ^Bb^	5.66 ± 0.038 ^Ba^
T_2_	7.68 ± 0.042 ^c^	7.40 ± 0.036 ^Ab^	6.62 ± 0.040 ^Aa^
Texture			
C	7.62 ± 0.024 ^c^	6.50 ± 0.021 ^Bb^	5.65 ± 0.028 ^Ba^
T_1_	7.65 ± 0.026 ^c^	6.57 ± 0.024 ^Bb^	5.71 ± 0.022 ^Ba^
T_2_	7.69 ± 0.028 ^c^	7.43 ± 0.025 ^Ab^	6.65 ± 0.026 ^Aa^
Juiciness			
C	7.61 ± 0.066 ^c^	6.40 ± 0.060 ^Bb^	5.55 ± 0.058 ^Ba^
T_1_	7.64 ± 0.042 ^c^	7.48 ± 0.044 ^Ab^	6.64 ± 0.040 ^Aa^
T_2_	7.66 ± 0.033 ^c^	7.55 ± 0.032 ^Ab^	6.74 ± 0.038 ^Aa^
Overall acceptability			
C	7.40 ± 0.038 ^c^	6.50 ± 0.034 ^Bb^	5.40 ± 0.036 ^Ba^
T1	7.45 ± 0.040 ^c^	6.57 ± 0.039 ^Bb^	5.46 ± 0.034 ^Ba^
T2	7.49 ± 0.040 ^c^	7.46 ± 0.040 ^Ab^	6.85 ± 0.043 ^Aa^

Mean ± SE with different superscripts in a row (lower case alphabet) and column (upper case alphabet) differ significantly (*p* < 0.05). 8-point descriptive scale was used (1 = liked extremely and 8 = disliked extremely). In total, 10 trained panelists evaluated the samples thrice (for each treatment). C = Samples without film, T_1_ = Samples with control film without *A. vera,* T_2_ = samples with a film containing 1.0% *A. vera* extract.

## Data Availability

The data are available from the corresponding author.

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
