# Peer review of "Aloe barbadensis Based Bioactive Edible Film Improved Lipid Stability and Microbial Quality of the Cheese"

_foods, 2023, doi:10.3390/foods12020229_

Round 1
Reviewer 1 Report
The manuscript showed a potential of Aloe for shelf-life extension packaging. However, there are only 2 treatments which are too few to show novelty. The discussion is still lack of novelty though tthey look good.
There shoulf be more literature about Aloe and packaging
Materials and method -> Add country to all chemicals and instrument. Add number of replications in all experiments.
Results -> Treatment are too few and most of them are somewhat as expected. It seems only the proof of the presence of Aloe that will give benefit to food quality.
Author Response
Responses to Reviewers comments:
The authors are highly thankful to the reviewers for their worthy suggestions that has improved the quality of the manuscript. We appreciate the efforts of editorial office for their time for reviewing the manuscript. The manuscript has been revised as per the suggestions of the reviewers and editor and the changes done/justifications provided are indicated by the red coloured text.
Reviewer 1
The manuscript showed a potential of Aloe for shelf-life extension packaging. However, there are only 2 treatments which are too few to show novelty. The discussion is still lack of novelty though they look good. There should be more literature about Aloe and packaging.
Done, have added some literature on Aloe and packaging, see line no. 29-44 and 58-67.
Preliminary trials were conducted to optimise the level of incorporation of A. vera gel in the film. Based on the results, 1% was found optimum. Addition of more than this level affected the quality of the film and therefore only two treatments were selected for the study along with the control. The standardization part is submitted to other journal for publication.
Materials and method -> Add country to all chemicals and instrument. Add number of replications in all experiments. Results -> Treatment are too few and most of them are somewhat as expected. It seems only the proof of the presence of Aloe that will give benefit to food quality.
Done, added countries of origin for chemicals and equipment, see materials and methods, red coloured text.
Done, added the replications below each of the tables and figures and also in the statistical section (n = 6).
Reviewer 2 Report
Thanks to submit "Aloe barbadensis Based Bioactive Edible Film Improved Lipid Stability and Microbial Quality of the Cheese" to Foods
One of the weaknesses of the work is that only one concentration of aloe vera has been tested. if more concentrations had been tested, perhaps the same bioactivity of the film would have been found decreasing the impact of the Aloe vera extract on the physical characteristics of the film.
Also, why was carrageenan chosen?
Aloe vera in italic. Correct in whole text.
Line#14: italic. how so great? this sentence does not need to be in the summary.
Line#16: italic
Line#47: italic
Line#47-49: I don't think it is necessary to say in the introduction which analyzes will be carried out in the work?.
Line#192: identify how many judges of each gender
Line#196: Add more information about team training
Line#198: Add Research Ethics Committee authorization
Figures: as there are only two treatments for quick reading I suggest putting the names as control and Aloe vera
Author Response
Responses to Reviewers comments:
The authors are highly thankful to the reviewers for their worthy suggestions that has improved the quality of the manuscript. We appreciate the efforts of editorial office for their time for reviewing the manuscript. The manuscript has been revised as per the suggestions of the reviewers and the changes done/justifications provided are indicated by the red coloured text.
Reviewer 2:
Thanks to submit "Aloe barbadensis Based Bioactive Edible Film Improved Lipid Stability and Microbial Quality of the Cheese" to Foods
One of the weaknesses of the work is that only one concentration of aloe vera has been tested. if more concentrations had been tested, perhaps the same bioactivity of the film would have been found decreasing the impact of the Aloe vera extract on the physical characteristics of the film. Also, why was carrageenan chosen?
Thank you for your comments. Preliminary trials were conducted to standardise the optimum level of incorporation of A. vera gel in the film. Different hydrocolloids were tried such as carrageenan, sodium alginate and some proteins. Based on the results, 1% A. vera and carrageenan were found optimum for the film. Addition of more than 1% level affected the quality of the film and therefore only two treatments were selected for the study along with the control. This level (1%) resulted in a film with suitable film characteristics and good antioxidant and antimicrobial properties. The standardization part is submitted to other journal for publication.
Aloe vera in italic. Correct in whole text.
Done, have corrected it throughout the manuscript.
Line#14: italic
Done, see line no. 14
Line#16: italic
Done, see line no. 16.
Line#47: italic
Done, see line no. 47
Line#47-49: I don't think it is necessary to say in the introduction which analyzes will be carried out in the work?.
Done, deleted the part from the introduction.
Line#192: identify how many judges of each gender
Done, see the section on sensory.
Line#196: Add more information about team training
Done, see the section on sensory.
Line#198: Add Research Ethics Committee authorization
Done, see sensory section.
No Research Ethics Committee authorization is required for sensory analysis in India.
Figures: as there are only two treatments for quick reading, I suggest putting the names as control and Aloe vera
Done, revised the figures as suggested, see figures.
Round 2
Reviewer 1 Report
-